# tRNA-Derived Small RNAs: Biogenesis, Modification, Function and Potential Impact on Human Disease Development

**DOI:** 10.3390/genes9120607

**Published:** 2018-12-05

**Authors:** Vera Oberbauer, Matthias R. Schaefer

**Affiliations:** Division of Cell and Developmental Biology, Center for Anatomy and Cell Biology, Medical University Vienna, Schwarzspanierstrasse 17, A-1090 Vienna, Austria; vera.oberbauer@meduniwien.ac.at

**Keywords:** tRNA, small RNAs, tRNA fragment, RNA modifications, protein translation, human disease

## Abstract

Transfer RNAs (tRNAs) are abundant small non-coding RNAs that are crucially important for decoding genetic information. Besides fulfilling canonical roles as adaptor molecules during protein synthesis, tRNAs are also the source of a heterogeneous class of small RNAs, tRNA-derived small RNAs (tsRNAs). Occurrence and the relatively high abundance of tsRNAs has been noted in many high-throughput sequencing data sets, leading to largely correlative assumptions about their potential as biologically active entities. tRNAs are also the most modified RNAs in any cell type. Mutations in tRNA biogenesis factors including tRNA modification enzymes correlate with a variety of human disease syndromes. However, whether it is the lack of tRNAs or the activity of functionally relevant tsRNAs that are causative for human disease development remains to be elucidated. Here, we review the current knowledge in regard to tsRNAs biogenesis, including the impact of RNA modifications on tRNA stability and discuss the existing experimental evidence in support for the seemingly large functional spectrum being proposed for tsRNAs. We also argue that improved methodology allowing exact quantification and specific manipulation of tsRNAs will be necessary before developing these small RNAs into diagnostic biomarkers and when aiming to harness them for therapeutic purposes.

## 1. Introduction

Transfer RNAs (tRNAs) represent the second most abundant RNA species in all cells playing a central role in the decoding of messenger RNA (mRNA) during protein translation. Besides this canonical function, tRNAs are also involved in numerous additional cellular pathways and metabolic processes. Some of these functions, such as providing aminoacyl-tRNAs for conjugation during protein degradation via N-end rule pathways or tRNA contribution to specific signal transduction, are common to both prokaryotes and eukaryotes (reviewed in References [1,2,3,4]). In contrast, other processes might be specific for only prokaryotic cells, such as the use of tRNAs as amino acid donors for cell wall biosynthesis (reviewed in Reference [5]), or have only been observed in eukaryotic cells, such as tRNAs affecting the modulation of apoptosis [6] or the use of tRNAs as primers during retroviral replication [7].

Importantly, both eukaryotic and prokaryotic tRNAs can give rise to various small RNAs, which appear to be not just remnants of tRNA maturation processes, nor are they mere tRNA degradation products. This heterogeneous population of small RNAs, called ‘tRNA-derived small RNAs’ (tsRNAs, for reasons of nomenclature inconsistency in published literature) throughout this review, represent rather distinct and stable entities potentially serving biological roles beyond those of their parental molecules.

Transfer RNA restriction systems in bacteria have been known for a quite some time [8]. In addition, various eukaryotic ribotoxins attacking tRNAs (among other RNAs) have also been discovered [9,10,11]. Rather recently, stress-induced and endonuclease-mediated tRNA fragmentation (resulting in small RNAs named tsRNAs^S^ in this review) as well as constitutive, tissue- and even cell type-specific tRNA fragmentation (yielding small RNAs named tsRNAs^NonS^ in this review) has been reported [12,13] (Figure 1). While tsRNA production and function has been implicated in a wide range of biological pathways, the mechanistic underpinnings as to how exactly individual tsRNAs would impact specific cellular processes remains largely unclear. Importantly, whether or not tsRNAs function alone or act through association with specific proteins (or other RNAs) and if post-transcriptional RNA modifications would affect such interactions is unknown. This review summarizes the current knowledge as to how different tsRNAs are produced, which potential functions they could have and which functions have actually been experimentally proven.

## 2. tRNAs: Ancient Molecules Serving Multiple Functions

Comparative sequence analysis of tRNA suggested that one-third of the present tRNA sequence divergence was already present at the time when archaea separated from bacteria [14]. Importantly, our primordial tRNA ancestors co-evolved with aminoacyl-tRNA-synthetase (aaRS) genes allowing the transition from non-ribosomal protein synthesis (by thioester formation) to an interplay of tRNAs, aaRSs and ribosomes (using phosphate ester formation) as the basis of ‘modern’ protein translation [15]. Since tRNAs are such ancient ‘molecular tools’ it is rather likely that their (multi-)functionality has been extensively tested throughout evolution resulting in additional ‘usability modes’ beyond their primary role in protein biosynthesis.

### 2.1. Protein Translation: The Canonical Function of tRNAs

tRNAs are indispensable components of the protein translation machinery. tRNAs display high conservation in function and structure. Cytoplasmic tRNAs are usually 73–90 nucleotides long while mitochondrial tRNAs (mt-tRNAs) can be as short as 57 nucleotides and still be functional. All tRNAs are characterized by the arrangement of a cloverleaf-like secondary structure with four main loops (D-, anticodon-, variable and T-loops). Mature tRNA undergoes coaxial stacking, especially of D- and T-loops forming the distinctive L-shaped tertiary structure [16]. The evolutionarily oldest and conserved function of all tRNAs is acting as adaptors during protein synthesis [17]. To execute their role in translation, tRNAs are charged with cognate amino acids through the activity of aaRS resulting in aminoacyl-tRNAs (aa-tRNAs), which decode mRNAs in the ribosomal P-site [18].

### 2.2. There is More to tRNAs: (Known) Non-Canonical tRNA Functions

Over the past 40 years, various non-canonical functions of tRNAs have been discovered. For example, specific tRNAs participate in bacterial cell wall biogenesis. To do so, the cell wall component peptidoglycan, a polymer substituted with peptides of common structure, is frequently crosslinked by tRNA-dependent aminoacyl-ligases [19]. Furthermore, bacteria use aa-tRNAs for the biogenesis of particular antibiotics [20,21]. Moreover, bacterial aa-tRNAs are also involved in the aminoacylation of membrane lipids. As a result, positive charge is introduced to the surface of bacterial membranes resulting in reduced affinity to cationic antimicrobial peptides (CAMPs) and thereby conferring resistance to antimicrobial peptides [22]. Importantly, both pro- and eukaryotes use aa-tRNAs for protein degradation. The mechanism was first described in *Saccharomyces cerevisiae* and called the N-end rule [23]. Here, a primary destabilizing amino acid from an aa-tRNA substrate is transferred to the N-terminal residue of a polypeptide thereby labelling it as substrate for the cellular protein degradation machinery [24].

In addition, non-aminoacylated tRNAs do participate in diverse processes tailored to cell survival. For instance, eukaryotic cells experiencing amino acid starvation reduce aminoacylation levels of tRNAs resulting in signalling through particular sensor aaRS and global gene expression changes [25,26]. Furthermore, tRNAs also modulated cytochrome C-mediated caspase activation thereby blocking apoptosis initiation [6] indicating that tRNAs can also be used as signalling adapters with a broad impact on cellular physiology. These combined findings suggest that tRNAs are indeed multi-functional molecules that have been adapted for a variety of functions, all of which might not have been discovered yet.

## 3. tRNA Biogenesis and Function Depends on Chemical Modifications

Transcription of pre-tRNAs is succeeded by an intricate and coordinated tRNA maturation process. End-processing (removal of 5’-leaders and 3’-trailers), successive nucleotide modifications, tRNA splicing and aminoacylation are all quality control steps in the production of mature and functional tRNAs. Numerous enzymes are involved in these processes including RNA exo- and endonucleases, RNA ligases and RNA modification enzymes [13].

### 3.1. tRNAs Are Highly Modified for Many Reasons

More than 150 post-transcriptional modifications have been described in tRNAs [27]. Up to 20% of all nucleotides in tRNAs can carry modifications. Importantly, chemical tRNA modifications and isomerizations are found in all characterized tRNA species and are conserved within each domain of life indicating their importance for both tRNA maturation and function [28]. Specifically, RNA modifications prevent degradation of pre- and even mature tRNAs [29], they modulate the efficiency and specificity of amino acid charging [30,31] and alter the specificity and decoding frame during protein translation as well as affecting translational speed [32]. Generally, ribonucleotide modifications in tRNAs conform to three rules: (1) modifications at different positions specifically affect tRNA identity and ensure proper acylation, (2) modifications in the main body of tRNAs affect folding and stability and (3) modifications in or around the anticodon loop are responsible for correct codon usage and affect translation (reviewed in Reference [13]).

### 3.2. RNA Modification Systems and the ‘Epitranscriptome’: Reversible or Not?

Specific RNA modifications are catalysed by a set of enzymes (‘writers’), which can be interpreted by specific proteins (‘readers’). The discovery of enzymatic activities that can remove particular RNA modifications (i.e., ‘erasers’ for adenosine methylations) has led to the coining of the term ‘epitranscriptome’ implying the existence of a dynamic network of RNA modification writers, readers and erasers, in close analogy to epigenetic modification systems. However, the defining characteristics of reversibility has only been reported for RNAs (including tRNAs) containing m^1^A, m^6^A and m^6^Am [33,34,35,36,37,38]. Of note, reversible modifications display a common chemistry, namely a nitrogen-carbon bond between modifier group and base that can be broken by, for instance, oxidative demethylation. Demethylation reactions are catalysed by Fe^2+^- and α-ketoglutarate-dependent dioxygenases, such as fat-mass and obesity (FTO)-associated protein and ALKBH5 enzymes (reviewed in Reference [39]). In contrast, modifications of different chemical composition such as carbon-carbon formation in m^5^C or pseudo-uridinylation, while being dynamically installed, are thought not to be readily reversible (discussed in Reference [40]). However, the recently reported in vivo C–C bond cleavage of 5′-formylcytosine resulting in cytosine in DNA suggested the possibility that further modification of the chemical group could allow reversibility under particular conditions and in specific cell types, potentially even in RNA [41]. Consequently, these issues raise the question as to how semantically distinguish RNA modification systems from one another. We suggest strictly calling only RNA modifications that are proven to be reversible as ‘epitranscriptomic’, while subsuming all others under the term RNA modifications.

## 4. RNA Modifications and Their Impact on tRNA Stability

The existence of sequence-specific DNA modification/restriction systems regulating endonucleolytic cleavage of genomic DNA are well known from bacterial systems. Because tRNAs are the most highly modified RNAs and since tsRNAs are produced through the activity of various endonucleases attacking single-stranded RNA (ssRNA) in open loop structures, the question arises as to what extent chemical modifications could modulate endonucleolytic cleavage of mature tRNAs thereby influencing tsRNA biogenesis.

### 4.1. Specific tRNA Modifications Attract and Repel Endonuclease Activity

Various bacterial and fungal endonucleases (ribotoxins) depend on specific nucleotide modifications in the anticodon loop of their target tRNAs. For example, a hyper-modification of guanine (queuosine, Q) determined the activity of Colicin E5 RNase at Q34 of four tRNAs in *Escherichia coli* [42]. Furthermore, a modified adenosine at position 37 (t^6^A37) or 5-methylaminomethyl-2-thiouridine at position 34 (mnm^5^s^2^U34) in tRNA-Lys enhanced the cleavage activity of bacterial PrrC, while pseudo-uridine at position 39 (Ψ39) was inhibitory [43]. The modified side chain at uridine 34 (mcm^5^s^2^U34) was also important for targeting fungal zymocin to tRNAs [44]. In contrast, PaT—a ribotoxin in *Pichia acacia*—specifically cleaved 3’ of mcm^5^s^2^U34 in the anticodon loop of tRNA-Gln^UUG^ but unlike zymocin, this cleavage occurred independently of mcm^5^s^2^U34 [45].

Since many tRNA modifications support various ‘proofreading’ steps during tRNA maturation into a functional cloverleaf structure, loss of particular tRNA modifications might impact efficient folding thereby allowing stress-induced endonucleases to better access ssRNA regions. To what extent single tRNA modifications can impact tRNA folding is exemplified by the findings that the positioning of m^1^A at A9 shifts the equilibrium between two major structures of tRNA-Lys [46]. Supporting this notion, cytosine-5 methylation (m^5^C), catalysed by Dnmt2/Trdmt1 at position 38, modulated stress-induced tRNA cleavage of tRNA-Asp^GUC^, tRNA-Gly^GCC^ and tRNA-Val^AAC^ in *Drosophila melanogaster* and suppressed ANG-mediated cleavage of these tRNAs when extracted from mouse cells [47]. Interestingly, m^5^C at C38 in tRNA-Asp^GUC^ is modulated by the presence of Q, a guanosine modification at G34, which has been detected in four tRNAs [48,49,50] and Q affected ANG-dependent tRNA cleavage in vitro [51]. Furthermore, the activity of another tRNA modification enzyme, NSun2, which targets m^5^C to most tRNAs, was also required for tRNA stability [52,53]. These observations suggest that specific RNA modifications likely change the structural context of tRNAs thereby modulating access for endonuclease activities or serving as platforms for specific enzymes.

### 4.2. tRNA Modifications Protect from Exonuclease Activities

The fact that tsRNAs are detectable as specific small RNAs raises the question as to how they evade exonucleolytic degradation after tRNA cleavage. The stability of small RNA species relies on various factors including nucleotide modifications, hybridization with complementary RNAs as well as association with effector proteins [54]. Importantly, tsRNAs in human serum appeared to be devoid of proteins and recent findings introduced the possibility of cross-hybridization between tsRNAs from different parental isoacceptors [55] suggesting that such dsRNA structures confer resistance to nuclease attack. Furthermore, 2′-*O*-methylation at 3′ ends, mediated by RNA methyltransferase *Hen1* homologues, is a conserved mechanism protecting various small RNA species against 3′-5′ exonucleolytic degradation [56,57,58,59,60,61]. In contrast, 3′ uridinylation targets small RNAs towards degradation [62]. Interestingly, tRNAs bear 2′-*O*-methylated nucleotides (Cm, Gm, Um), not at 3′ ends but located in D-, variable and anticodon loops, which is in close proximity to mapped endonucleolytic cleavage sites. These modified nucleotides could limit the extent of exonuclease activity and thus increase the stability of produced tsRNAs. Indeed, tsRNAs^NonS^ originating from tRNA-Gln^GUG^ were ribose 2′-*O*-methylated at 3′ ends [63]. In this context it is important to mention that particular nucleotide modifications not only improved the stability of synthetic small RNAs [64] but also reduced the antigenicity of tRNA sequences [65,66] suggesting that a combination of various modifications in tsRNAs might contribute to their stability and functional impact (Figure 2).

## 5. tsRNAs: Degradation Leftovers or Functional Small RNAs?

Enzymes that can cleave tRNAs and even tRNA fragments have been described in the 1970s [67,68]. Around the same time, tRNA-derived degradation products in the form of modified nucleosides were detected in urine from tumour patients and interpreted to be a result of high tRNA turnover in cancer cells [69,70]. However, only recently it became possible to detect and catalogue tsRNAs as reproducible entities in many cell types and tissues exposed to varying environmental conditions. In the following sections, we will briefly describe the various identities of these tsRNAs, introduce nuclease activities responsible for their production and mention the conditions and biological settings leading to tsRNA production.

### 5.1. tsRNA-Nomenclature: Let’s Try Giving ‘Them’ Names!

Since the beginning of this century, tsRNAs have been recognized as small RNAs with the potential to be biologically relevant. However, categorization and naming of tsRNAs, both in literature and public databases, is still largely inconsistent [71,72,73].

Starvation-induced tsRNAs^S^ in eukaryotic cells were first reported in *Tetrahymena* and simply named tRNA fragments [12]. However, other authors chose to label the products of stress-induced tRNA fragmentation in different organisms with different names such as stress-induced tRNA-derived RNAs (sitRNAs, [74]) or tRNA-derived, stress-induced RNAs (tiRNAs, [75]). In addition, tsRNAs existing under steady-state conditions were either called tRNA-derived fragments (tRFs [76]) or tRNA-derived small RNAs (tsRNAs, [77]). Others have called these RNAs, tRNA-derived RNAs (tDRs, [78]) or even tRNA-derived small non-coding RNA (tsncRNAs, [79]).

Although various authors attempted introducing some order into the nomenclature, the net result was only a distinction between tRNA halves (28–36 nucleotides) that are largely stress-induced (often also called tiRNAs) and tRFs (14–30 nucleotides) that are produced under steady state conditions. We will call the former tsRNAs^S^ and the latter tsRNAs^NonS^ throughout this review (Figure 1). Unfortunately, the term ‘tiRNA’ remains ambiguous since it has also been used to name ‘transcription initiation RNAs’ [80,81]. Therefore, attempts to systematically name tsRNAs and assign recognizable nomenclature for reproducible future use are very welcome [82].

### 5.2. Processing and Maturation: Nucleases Acting on tRNAs

tRNAs can be extremely stable with half-lives measured in days and this stability is connected to numerous nucleotide modifications as well as to correct tertiary structure. Aberrantly processed or hypo-modified tRNAs become destabilized and are rapidly degraded by various exonucleolytic activities that are part of two molecular pathways: the TRAMP/nuclear exosome and the ‘rapid tRNA decay’ (reviewed in Reference [13]). The existence of such RNA degradation pathways indicated the need for cells to sense and remove hypo-modified tRNAs [83]. Importantly, dysfunctional RNA surveillance can cause imbalances in small RNA pathways, as reported in a study in *Schizosaccharomyces pombe* that demonstrated that TRAMP-mediated RNA degradation prevents entry of tRNAs and other RNAs into the siRNA pathway [84]. Whether by-products of these constitutively active tRNA degradation pathways give rise to defined tsRNAs^NonS^ remains to be tested.

On the other hand, correct tRNA maturation is a multistep process involving the removal of pre-tRNA sequences by RNaseP and RNaseZ-like enzymes (ca. 50 nucleotides at the 5′ and 3′ end, respectively). In addition, various tRNAs contain introns that are ‘spliced’ during their maturation process. tRNA splicing endonuclease activities create nicked tRNA precursors, which are substrates for the tRNA ligase complex creating mature tRNAs. Mutations in the latter complex resulted in the accumulation of tsRNAs^NonS^ representing non-ligated tRNA splice intermediates that were diagnostic for particular human disease syndromes [85,86,87].

In contrast to general tRNA processing, degradation and splicing mechanisms, various endonucleases target correctly formed and functional tRNAs, preferentially at open loop structures (D-, anticodon-, variable- and T-loops) resulting in ‘nicked’ tRNAs containing a 2′-3′-cyclic phosphate at the 3′-end and a 5′-OH at the 5′ end of the hydrolysed RNAs. Particular killer ribotoxins have first been described in phage-infected bacteria [88] and later also in simple eukaryotes [11,89]. ACNases cleave various tRNAs in the anticodon loop, causing cytotoxicity and cell death [8]. For instance, VapCs are site-specific endonucleases that cleave tRNA-fMet in the anticodon stem-loop in *Shigella flexneri* and *Salmonella enterica* thereby reprogramming translation [90]. Furthermore, *E. coli* endonuclease PrrC cleaves tRNA-Lys^UUU^ in a suicidal attempt to defend the population against infection by bacteriophage T4 [45]. Furthermore, the actively secreted bacterial plasmid-encoded ACNases colicin D and E5 cleave tRNA-Tyr^QUA^, tRNA-His^QUG^, tRNA-Asn^QUU^, tRNA-Asp^QUC^ and tRNA-Arg^ICG/CCG/U*CU/CCU^ (Q indicates queuosine, U* indicates the modification m^5^U) in order to defend the strain against competing *E. coli* [91,92]. Both nucleases, when expressed in eukaryotic cells, were toxic [93,94], which suggested also susceptibility of eukaryotic cells to tRNA fragmentation. Indeed, fungi such as *Kluyveromyces lactis* and *Pichia acaciae* produce ACNases (zymocin and PaT, respectively), which cleave specific tRNAs in the anticodon loop as a defence against non-self species [44,95].

Of note, vertebrates also express cytotoxic tRNA endonucleases, namely Angiogenin (ANG), onconase (Ranpirnase) and amphinase (Amph) [11,96,97]. Recently, RNaseL, an interferon-inducible endonuclease, has been implicated in targeting tRNAs thereby producing tsRNAs [98]. While ANG is clearly an ACNase, Ranpirnase cleaves tRNAs predominantly in the variable loop regardless of the sequence context [99] and Amph and RNaseL appear to have an even wider RNA substrate specificity. ANG is an RNase A-family enzyme that is internalized via receptor-mediated endocytosis by binding to Plexin-B2 [100], while Ranpirnase enters cells in a manner similar to cell-penetrating peptides using an abundance of lysine residues [101].

The biological role for such secreted cytotoxic endonucleases in multicellular organisms beyond defence against invasive agents is not completely clear. Of note, ANG is highly produced by liver cells and secreted into the bloodstream [102,103]. ANG promotes blood vessel growth [104,105] and neuronal cell survival [106]. Particular isoforms of ANG display anti-microbial properties suggesting activities on microbial tRNAs [107]. Interestingly, both Ranpirnase and Amph display antiviral and anti-tumour activities, which suggested additional substrates beyond tRNAs for these enzymes [99,108,109], while ANG has the opposite effect through its angiogenic activities thereby promoting tumour growth [110].

### 5.3. Environmental Stress: tRNA Fragmentation as Conserved Cellular Response

The exploration of fungal ribotoxins has been a major driver for elucidating the mechanisms, structures and evolution of eukaryal tRNA restriction enzymes, especially in light of the recent discoveries that tRNA cleavage is a general response to stress. Two eukaryotic ACNase protein families (RNase A and T2) specifically cleave mature tRNAs in response to stress. Of note, Rny1 (RNaseT2) is an ACNase in *S. cerevisiae* [111] while other eukaryotic RNaseT2 family members appear to target tRNAs in every loop structure [112]. In contrast, the RNaseA-family member Angiogenin attacks many tRNAs during the stress response in mammalian cells [75]. Both ACNase family members are normally sequestered (and inhibited by binding to proteins) from cytoplasmic tRNAs and only become activated upon stress exposure. The production of distinct tsRNAs^S^ has been detected after starvation [12], oxidative stress [75,113,114], nutritional deficiency [77], hypoxia and hypothermia [115,116], heat shock or ultraviolet irradiation [74,75,117]. In addition, tsRNAs^S^ appeared to increase during aging. A study in *Caenorhabditis elegans* reported elevated levels of tsRNAs in an age-dependent manner indicating increased stress responses in older animals [118]. However, especially in whole organisms it is often impossible to clearly distinguish whether tsRNAs^S^ are produced in response to a particular stress or as part of a developmental process exerting stress on certain cell types.

### 5.4. Developmental tRNA Fragmentation

tsRNAs can also be generated in the absence of overt stress exposure. In *Streptomyces coelicolor*, changes in tRNA fragmentation were observed during vegetative growth, aerial hyphae formation and sporulation. In more detail, increased levels of tsRNAs^NonS^ were detectable at the initiation of aerial hyphae formation followed by a reduction after bacterial development had proceeded through sporulation [119]. tsRNAs^NonS^ were also observed upon conidiation of *Aspergillus fumigatus* resulting in depletion of full length tRNAs in the conidia, which suggested that tRNA fragmentation might serve to stall protein synthesis of filamentous fungi during their resting state [120]. These observations indicated a role for active tRNA fragmentation during bacterial development and particular life cycle stages.

Stress-independent tsRNAs have also been observed in mammals. For instance, high levels of tsRNAs^NonS^ were detected in both hematopoietic and lymphoid systems as well as in cell-and vesicle-free blood fractions suggesting that tsRNAs can be secreted and exist as stable entities outside of membranous organelles [55,121,122,123]. In contrast, others reported tsRNA^NonS^ content in extracellular vesicles [124] and comparison of cellular and vesicle-borne tsRNAs in the immune system showed that immune cells contained exclusively 30–35 nt-long tsRNAs while extracellular vesicles carried 40–50 nt-long tsRNAs [125]. These combined observations suggested that specific tsRNAs could serve as signalling molecules in blood and lymphatic circulatory systems. Interestingly, specific tsRNAs^S^ were also detected in human breast milk [126] and in mature mouse sperm [127,128] indicating tsRNA^S^ production for directed transmission into the next generation.

## 6. Biological Functions of tsRNAs: There Is a Lot of Potential

Various studies using hybridization techniques indicated that only 0.1–5% of a given tRNA isoacceptor yield tsRNAs, especially under stress conditions. This suggested that any amount of tsRNAs is likely very low if averaged over the RNA content of single cells or tissues. However, when taking into consideration the possibility that tsRNA production could occur rather locally within cells and thereby would remain confined to, for instance, particular subcellular ribonucleoprotein particles or membrane-less organelles (i.e., ribosomes, stress granules), the amount of locally produced tsRNAs might become extraordinarily high and thereby locally effective. We, therefore, suggest that any potential biological function of tsRNAs should be considered under the assumption that tsRNAs are produced locally in high relative concentrations.

Only very few experiments have been performed which unequivocally (e.g., by direct manipulation of tsRNAs) tested the involvement of tsRNAs in the biological processes that were associated with their occurrence. This experimental conundrum is further complicated by the fact that it is often impossible to separate the function of specific tsRNAs from those of their parental tRNAs (intact or ‘nicked’) or from phenotypes caused by mutations in tRNA processing and modification enzymes. In this respect, the use of antisense oligomer-mediated knock-down with the aim of targeting tsRNAs but not tRNAs remains questionable even though interference with full length tRNAs were reported to be absent [76,129,130,131,132]. When considering that only fewer than 5% of all molecules of a given tRNA isoacceptor will be targeted to produce tsRNAs (i.e., during stress conditions), it remains unclear how antisense oligonucleotides seek out those few tsRNAs instead of the majority of parental tRNA molecules. In addition, absolute quantification of tRNAs and tsRNAs using next-generation-sequencing (NGS)-based techniques should be critically assessed because the modification landscape of any tRNA-derived molecule will greatly affect all reverse transcriptase (RT)-based assays thereby shifting readouts in unpredictable ways [133]. 

To facilitate orientation, we introduce here an experimental ‘categorization’, which might allow better judging the existing experimental evidence for tsRNA involvement in a particular process (see Table 1, Figure 3).

### 6.1. tsRNAs Interfere with Protein Translation

Particular 5′-tsRNAs^S^ directly interfered with translation [75,114,136,137,139,151,152,155,156]. As one mode of molecular action, specific tsRNAs^S^ displaced various eukaryotic initiation factors from both capped and uncapped mRNA causing translational repression [114]. Two particular isoacceptor-specific tsRNAs^S^ were most potent in causing translational repression, namely tsRNA-Ala^UGC^ and tsRNA-Cys^GCA^ [156]. These tsRNAs^S^ were comprised of 5′ halves sharing two structural features: the D-loop, supposedly binding translational repressor proteins and a terminal oligo-guanine (TOG) motif [114]. TOG motif interactions facilitated RNA G-quadruplex formation (RG4s) involving four individual 5′-tsRNAs^S^. Interestingly, RG4s were required for the initiation of SGs triggered by tsRNAs^S^ in vivo [152].

In another mode of action, tsRNAs^S^ indirectly interfered with protein synthesis, in particular through binding to ribosomal components. For instance, specific 5′-tsRNAs^S^ (tsRNA-Val^GAC^) resulting from tRNA cleavage in or around the D-loop in the archaeon *Haloferax volcanii* interacted with the small ribosomal subunit [117]. Interestingly, these tsRNAs competed with mRNA for ribosomal binding in response to alkaline stress conditions. Importantly, a similar interaction between tsRNAs and ribosomes without the need for complementarity to mRNAs was reported for eukaryotes and prokaryotes, indicating a conservation of this mechanism during stress conditions [139,151].

However, a recent report suggested that tsRNAs do exactly the opposite of interfering with protein translation. In this case, a 3′ tsRNA-Leu^CAG^ enhanced the translation of two ribosomal proteins, which are important for ribosomal biogenesis [132]. These combined observations suggest fragment-specific effects on or condition-dependent usage of tsRNAs during protein translation.

### 6.2. Can tsRNAs Act Like or Mimick Canonical Small RNAs?

Of note, tsRNAs were often wrongly annotated as micro RNAs (miRNAs) in databases [157]. Importantly, particular tsRNAs^NonS^ were detectable almost exclusively in the cytoplasm [76,158] and showed Dicer-dependent biogenesis [77,135]. However, various assays using sensor transcripts with complementary regions to tsRNAs^NonS^ did not reveal canonical miRNA or siRNA-like activity [76,77] indicating that it was unlikely that tsRNAs played direct regulatory roles in post-transcriptional gene silencing (PTGS). Instead, alternative effects on PTGS were suggested involving competitive association of tsRNAs^NonS^ with Argonaute (AGO) clade proteins [77]. Indeed, various data sets detected tsRNAs in AGO complexes [63,159,160,161,162] but it remained unclear whether this association could be biologically significant. Only recently, reports emerged on specific tsRNAs acting in a truly microRNA-like fashion. For instance, particular 5′-tsRNAs^NonS^ originating from Dicer-like 1 (DCL1) processing became incorporated into Argonaute 1 (AGO1) and accumulated in pollen of *Arabidopsis thaliana*. These tsRNAs^NonS^ specifically targeted mRNAs produced from transcriptionally active transposable elements [149]. tsRNAs^NonS^ production and loading into AGO1 and AGO2 in *Drosophila melanogaster* was age-dependent and bioinformatic prediction of potential target genes suggested regulatory roles in neural processes [144]. In addition, tsRNA mimicry of miRNAs was also reported in human cells. Here, pre-tRNA-Ile^UAU^ was recognized as miRNA precursor by exportin-5 (Xpo5) and transported to the cytoplasm, processed by Dicer and loaded into AGO effector proteins [146]. Furthermore, an abundant and Dicer-dependent tsRNA^NonS^ (CU1276) was also described in germinal centre-derived B cells but not in germinal centre-derived lymphomas. CU1276 was derived from tRNA-Gly^GCC^ and induced repression of the RPA1 gene in a miRNA-like fashion resulting in changes in DNA damage responses [140].

Lastly, tsRNAs also feature in the tug-of-war between viruses and host systems. For instance, tsRNA^NonS^ (tRF-3019) of host cells bound to the primer-binding sites (PBS) of retroviral RNA from the human T-cell leukaemia virus type 1 (HTLV-1) thereby initiating reverse transcription and promoting virus amplification [143]. In addition, respiratory syncytial virus (RSV) utilized host tsRNA-Glu^CUC^ to suppress mRNA expression of the antiviral apolipoprotein E receptor 2 (APOE-R2) by targeting its 3′ untranslated region in a miRNA-like fashion resulting in the amplification of RSV [145]. These combined observations suggest that the uncontrolled production of tsRNAs could be ‘double-sided sword’. tsRNAs could either act as small RNA mimics thereby ‘distracting’ small RNA pathways or perform functions as actual small RNAs in PTGS.

### 6.3. Could tsRNAs Serve as Sink or Decoy for Specific Proteins?

For processing and maturation, tRNAs must interact with a large number of different proteins. In addition, when tRNAs contribute to decoding of mRNAs not only do various aaRSs associate with different tRNAs before proofreading tRNA identity [163] but specific translation initiation factors and many ribosomal proteins do also interact with tRNAs. Since tsRNAs originate from pre-tRNAs and mature tRNAs, they could bind to tRNA interacting proteins thereby competing with their parental molecules. It follows that tsRNA binding to particular proteins might result in a sink for these molecules resulting in the modulation of specific molecular pathways and thereby cellular physiology. Indeed, evidence for such binding modes exist. For instance, both tRNAs and tsRNAs^S^ can bind to cytochrome C both in vivo and in vitro. The balance of these interactions appears to become relevant during hyperosmotic stress conditions when specific tsRNAs^S^ bound to cytochrome C and interfered with apoptosome formation thereby attenuating cell death signalling [142].

Furthermore, it was demonstrated that tsRNAs^S^ originating from the 5′ but not 3′-end of mature tRNAs repressed protein synthesis. Mechanistically, particular tsRNAs with a propensity to form RG4s inhibited translation initiation through binding to specific components of the cap complex thereby interfering with eIF4A/G assembly at mRNAs with potential implications for the cellular stress response [114,137,152,153,156]. Similarly, specific tsRNAs^S^ formed during hypoxic stress conditions suppressed the stability of multiple mRNAs thereby interfering with metastasis and cancer proliferation [129]. Specifically, tsRNAs^S^ bound to YBX1, an RNA-binding protein, thereby displacing its stabilizing influence from the 3′ UTRs of oncogenic mRNAs. Furthermore, excessive production of small RNAs, for instance by ectopic siRNA production in some experimental RNAi settings or through dysfunctional TRAMP activity, can be lethal partially by disrupting other small RNA pathways [135] resulting, for instance, in increased tsRNA loading into AGO1 complexes [84]. Similarly, if lupus autoantigen (La), which ensures correct processing of pre-tRNA into mature tRNAs, was knocked down, cleavage of pre-tRNAs into tsRNAs increased significantly, resulting in tsRNAs entering the miRNA pathway [146]. Of note, loading of an essential *Tetrahymena* Piwi protein (Twi12) with tsRNAs^NonS^ was required for nuclear import of Twi12 and the activation of RNA processing by the exoribonuclease Xrn2 indicating aptamer function of tsRNAs [138,164]. Interestingly, also a human Piwi-like protein (Hiwi2), when overexpressed in human breast adenocarcinoma cells, preferentially interacted with tsRNAs, which the authors named tRNA-derived piRNAs [165], further supporting the notion that small RNA pathway components can associate with tsRNAs under specific conditions. Of note, mutations in Dnmt2, a tRNA modification enzyme, caused increased tRNA fragmentation in flies resulting in pleiotropic phenotypes including the deregulation of small RNA pathways [141]. These combined observations suggest the existence of intricate systems controlling tsRNA production, localization and removal thereby avoiding such interactions constitutively but at the same time allowing tsRNA binding to various proteins when required.

### 6.4. Could tsRNAs Act as (Stress) Signals within Organisms?

Analysis of tsRNAs^S^ in plants indicated functional roles as long-distance signals. Specifically, tsRNAs^S^ produced in the leaves of the pumpkin *Cucurbita maxima* entered the phloem, thereby accessing circulatory networks of the plant and suggesting the possibility of function at a distance [136]. In support of this notion, high levels of tsRNAs^S^ have been detected in *A. thaliana* roots compared to shoots after controlled phosphate starvation [166], whereas oxidative stress resulted in high levels of tsRNAs^S^ in flowers [113]. These observations led to the testable assumption that tsRNAs^S^ could serve as signalling molecules acting at a distance potentially ‘informing’ other cells and tissues thereby allowing the rest of the plant to adapt its metabolism in response to, for instance, nutrient availability, pathogen infection or other environmental stimuli.

As tsRNAs were also detected in circulatory systems of mammals both in cell-free as well as in vesicle fractions, it was suggested that tsRNAs could serve as signalling molecules [78,121,122,123,167]. For instance, during acute inflammation, increased levels of tsRNAs^S^ could be detected in the circulatory system [168]. Furthermore, particular tsRNAs^S^ derived from tRNA-Ala^UGC^ and containing a CCACCA sequence at the 3′ end were sufficient to activate the immune responses of Th1 and toxic T lymphocytes through Toll-like receptors [134]. These combined observations suggest the possibility of directed tsRNA secretion upon stimulation to act in a systemic context as information carriers and signalling molecules.

### 6.5. Can tsRNAs Carry Information Between Organisms?

Specific tsRNAs were also found in bodily fluids such as saliva, tears, urine and breast milk [126,169,170]. Interestingly, mature mouse sperm contained high levels of tsRNAs [127,128,131,147,171,172]. Various intergenerational experiments suggested a role for sperm-borne tsRNAs in relaying extra-chromosomal information in response to diet manipulation [131,147,172,173,174]. Specifically, in offspring of males that were fed on high-fat or low-protein diet the development of metabolic changes including insulin resistance and glucose intolerance were observed. As to how the few inherited tsRNAs affect the complexity of gene expression patterns remains to be elucidated [175]. Furthermore, metabolic labelling revealed that tsRNAs entered maturing sperm through micro-vesicles (called epididymosomes), which are secreted from somatic cells lining the epididymis [176]. Interestingly, particular tsRNA modifications (i.e., m^5^C, m^2^G, m^1^A) appeared to change upon exposure to high-fat diet indicating that they play regulatory roles during the transfer of small RNA-based extrachromosomal information [172]. These combined observations clearly indicated that modified tsRNAs can act as information carriers between two generations in mammals.

## 7. tRNA Modification Systems Impact Human Disease Development

Transfer RNAs are central to any cellular process because of their involvement in protein synthesis. It follows that interference with protein synthesis will likely cause malfunctioning of cells, tissues, organs and organisms. Indeed, tRNA mutations, overexpression of specific tRNAs, incomplete tRNA processing or missing tRNA modifications often result in biological consequences impinging on aberrant protein translation [177]. Of note, genetic mutations in various tRNA modification systems are frequently associated with human disorders, disproportionately affecting proliferation and metabolically active tissues such as muscle and nervous system (reviewed in References [178,179,180,181,182]) supporting the notion that chemical modifications have a major impact on small RNA function (see Table 2).

## 8. Are tsRNAs Causative Agents or Just Signs for Particular Human Disease States?

As of now, connections between tsRNA function and human health remain largely phenomenological, descriptive and therefore only tentative [79,82,241,242]. Importantly, the multiplicity of assigned tsRNA functions has often not been dissociated from potential effects caused by the dysfunction of their parental tRNAs. Furthermore, tRNA levels vary widely between different human tissues that are considered healthy and non-diseased [243]. However, when human-derived material was experimentally queried for tsRNA abundance and potential function, only a few studies used patient-derived material (i.e., biopsies, blood samples), while the majority utilized only proxies for human disease states, especially cancer cell culture systems, resulting overwhelmingly in reports that correlated changes in tsRNA abundance and identity with certain disease states. In this respect, it is important to note that the use of long-term cultured primary cancer-derived cells, while preferable when elucidating basic molecular biology, are changing constantly in culture both genetically as well as epigenetically, thereby accommodating to the various stresses they encounter [244,245,246]. Such changes certainly include molecular circuitry impinging on protein synthesis including tRNA expression and metabolism [247,248,249,250]. Whether increased tRNA levels in cancer-derived cells consequently results in an increase in tsRNAs remains unknown.

Despite these experimental shortcomings, quantitative and qualitative changes might allow developing particular tsRNAs into useful biomarkers, with the aim to report on and monitor the state and progression of a particular human disease or syndrome [181,251]. On the other hand, it still remains to be determined whether aberrant production (too much or too little) of tsRNAs can actively contribute to disease pathogenesis. Below, we introduce a number of examples in which the abundance and function of particular tsRNAs has been linked to cancer, infection, neurodegeneration and other pathological conditions.

### 8.1. Signs for Stress and Viral Infection

Persistently activated stress responses and increased inflammation due to infection result in the need of constant controlling and repairing cellular damage and are central to and promote disease pathogenesis. Changes in tsRNA abundance have been observed in many stress-exposed or virus-infected cell types or tissues, therefore making the existence of these small RNAs proxies for sustained stress levels.

For instance, RSV infection, the most common cause of bronchiolitis and pneumonia in infants and elderly people, led to overproduction of ANG-dependent tsRNA^S^, which repressed particular host mRNAs thereby promoting RSV replication [252]. These findings are supported by a study in cell culture showing that persistent infections with hepatitis B or C viruses (HBV and HCV, respectively) resulted in hepatic cirrhosis and hepatocellular carcinoma (HCC). Profiling of small RNAs in liver from human subjects with advanced hepatitis B or C and HCC revealed that tsRNAs^S^ were significantly increased in humans with chronic viral hepatitis [253]. Importantly, in HBV-associated HCC, tsRNAs^S^ abundance correlated with expression of ANG demonstrating that tsRNAs^S^ are highly abundant in chronically infected livers and in liver cancer tissues.

### 8.2. Links to Cancer

As cancer cells are highly proliferative, their demand for elevated protein synthesis requires adjustment of the translation machinery. Indeed, cancer cells produce increased levels of tRNAs, ribosomal RNAs and ribosomes when compared to non-cancerous cells. Cancer cells also feature high tRNA turnover [69]. Importantly, the angiogenic ACNase ANG is overexpressed in almost all cancer types (reviewed in Reference [254]). Of note, its ribonuclease activity is critical for promoting angiogenesis and inhibiting this activity significantly inhibited tumour formation in xenograft tumour models [255,256]. Whether increased tRNA levels in human cancer-derived cells consequently results in an increase in tsRNAs remains unknown.

Links to cellular overproliferation can be made through the activity of RNaseZ/ELAC2, a general tRNA biogenesis enzyme, which was responsible for the production of tsRNAs^NonS^ (tRF-1) [76] and that has been identified as a prostate cancer susceptibility gene [257] thereby indirectly connecting tsRNA biogenesis with a particular cancer type.

On the other hand, specific tsRNAs^S^, termed Sex HOrmone-dependent tRNA-derived RNAs (SHOT-RNAs), were highly expressed in oestrogen receptor (ER)-positive cells from luminal-type breast cancer patient tissues [130]. These SHOT-RNAs were produced by angiogenin-mediated anticodon cleavage in a sex hormone-dependent fashion. These findings indicated tRNA cleavage as a result of particular hormonal changes resulting in functional involvement in cell proliferation. Also, high abundance of tsRNA^NonS^ (tRNA-Leu^CAG^) was reported in human non-small cell lung cancer (NSCLC) patient-derived tissues [150]. A significant relationship between cancer stage and this tsRNA was reported in NSCLC-derived patient sera. Furthermore, inhibition of this tsRNA suppressed cell proliferation and impeded cell cycle progression indicating a supportive role of this particular tsRNA^NonS^ in proliferation. Interestingly, inhibition of specifically this tsRNA (tRNA-Leu^CAG^) induced apoptosis in a patient-derived orthotopic hepatocellular carcinoma model in mice [132]. Mechanistically, this tsRNA associated with at least two mRNAs coding for ribosomal proteins (RPS28 and RPS15) thereby positively affecting their translation. Furthermore, tsRNAs targeting translation initiation in embryonic stem cells were dysregulated in aggressive subtypes of human myelodysplastic syndromes indicating a function for tsRNAs in hematopoietic stem cell commitment and germ layer specification [153]. These combined observations suggest a link between tsRNA function and the highly proliferative phenotype observed in stem cells and tumours.

### 8.3. Links to Neurodegeneration

Genetic mutations in tRNA processing and tRNA modification systems are frequently associated with human disorders [178,180], many of which affect metabolically active tissues such as the nervous system (Table 2). Strikingly, mutations in ANG, the stress-induced endonuclease producing tsRNA^S^, were implicated in the pathogenesis of Amyotrophic Lateral Sclerosis (ALS), a still incurable neurodegenerative disease [258]. Importantly, a subset of ALS-associated ANG mutations was also discovered in patients suffering from Parkinson’s Disease (PD) [259]. Interestingly, most ALS/PD-associated ANG mutations are clustering in the RNase domain of the enzyme [260]. Furthermore, ANG-dependent tsRNAs, if transfected into cells, increased motoneuron survival upon stress exposure, suggesting that tsRNA^S^ can confer neuroprotection [156].

In addition, several studies linked cellular stress, tRNA modification networks and ANG-mediated production of tsRNAs^S^ to neurodevelopmental disorders. For instance, mutations in the (cytosine-5) RNA methyltransferase NSun2 have been genetically linked to human forms of intellectual disability (ID) and to Dubowitz-like syndrome [195,196,197,198]. On the molecular level, *NSun2* null mutations in mice resulted in the ectopic production of tsRNA^S^, increased stress responses and neuronal cell death [53]. While removing a pivotal tRNA methyltransferase, which also targets other RNAs [261,262], linked tRNA modifications and tRNA metabolism with aberrant nervous system development, the particular contribution of tsRNAs to both growth retardation and impaired intellectual development in patients carrying NSun2 gene mutations remains unclear.

Furthermore, tsRNAs^NonS^ derived from intron-containing tRNAs accumulated in animal models of pontocerebellar hypoplasia (PCH), a heterogeneous group of inherited human neurodegenerative disorders characterized by developmental and neuromuscular defects [85,86,87]. One of the causative mutations for PCH affected the function of the RNA kinase CLP1, which is important for tRNA splicing. On the cellular level, interference with CLP1 function led to the accumulation of unspliced pre-tRNAs and ultimately to depletion of mature tRNAs in patient-derived neurons. Interestingly, accumulating tsRNAs^NonS^ in CLP1 mutants were represented by linear introns and 5′-tsRNAs [85,86] and transfection of particular 5′-tsRNAs into patient-derived neurons resulted in reduced survival, especially under oxidative stress [87]. While the exact mechanistic details on the interplay between tRNA biogenesis factors, the resulting depletion of mature tRNAs and the accumulation of particular tsRNAs are yet to be elucidated, these studies clearly link aberrant tRNA metabolism with the aetiology of neurodegenerative syndromes.

### 8.4. Links to Metabolic Syndromes

By utilizing historical records, various correlative findings have been made indicating that environmental factors affected human paternal transmission of obesity and associated metabolic disorders in the past [263,264]. While the mechanisms of the transmission of such complex phenotypes in humans remain unclear, various studies using animal models supported the notion that mammalian metabolism can be influenced paternally through nutrition-induced signals, which impinge on epigenetic mechanisms in the fertilized zygote [265]. Especially, the role for RNAs in bookmarking and transmitting information about parental exposure to various stress conditions has been controversially discussed [266,267,268,269,270].

Recently, two studies indicated that tsRNA transmission into fertilized mouse zygotes contributed to establishing gene expression programs that recapitulated metabolic syndrome-like phenotypes seen in the paternal organism [131,147]. Specifically, particular tsRNAs exhibited changes in abundance in the sperm from mice exposed to high-fat diet (HFD) [147] or low-protein diet [131]. Interestingly, also changes in RNA modification patterns were observed [147]. Of note, injection of small RNAs (in the size range of tsRNAs extracted from HFD sperm) into fertilized zygotes resulted in gene expression changes of metabolic pathways and islets of the offspring [147] and RNAi-mediated knockdown of particular tsRNAs resulted in the upregulation of specific genes that contained sequences derived from endogenous retro-elements (MERVL) [131]. Importantly, synthetic tsRNA injection into fertilized zygotes did not produce such changes indicating that RNA modifications in tsRNAs are pivotal to the function of these small RNAs [147]. Furthermore, deletion of DNMT2/TRDMT1, a (cytosine-5) tRNA methyltransferase, abolished sperm-mediated transmission of HFD-induced metabolic disorders into offspring [172] indicating that m^5^C contributed to biological properties of tsRNAs. In addition, paternal exercise appeared to influence the abundance of small RNAs in sperm and their transmission of ‘metabolic memory’ into the next generation [173]. These findings support the notion of direct effects of modified tsRNAs on intergenerational inheritance of acquired traits. Whether human sperm also contain tsRNAs with similar potency or if tsRNAs contained in breast milk carry some form of information from mother to infant remains to be tested.

Loss-of-function mutations in TRMT10A, a (1-methylguanosine) tRNA methyltransferase, are a monogenic cause of early onset diabetes. Using induced pluripotent stem cell-derived pancreatic β-like cells from TRMT10A-deficient patients it was reported that TRMT10A deficiency induced oxidative stress resulting in increased apoptosis of pancreatic β-cells [154]. Importantly, tRNA hypomethylation at position G9 led to tRNA fragmentation of specific tRNAs and the resulting tsRNAs mediated the β-cell death by unknown mechanisms supporting an active role for tsRNAs in changing the susceptibility of particular cell types to environmental stress.

### 8.5. Links to Microbiome Dysregulation?

Recent studies revealed the potential role of small RNAs in interspecies and cross-domain interactions [271]. These ‘social RNAs’ could serve as gene silencing and molecules serving other functions in recipient cells across domains, a phenomenon named ‘cross-domain RNAi’ [272]. For instance, association between colorectal cancer, miRNA expression and the gut microbiota has been reported in humans [273].

Of note, mammals maintain intricate symbiotic relationships with other eukaryotes (i.e., fungi) and with a plethora of prokaryotic organisms (representing the microbiota), all of which also maintain diverse molecular machineries for the production of tsRNAs. Interestingly, specific tsRNAs secreted into human saliva affected the growth of a key oral commensal and opportunistic pathogen *Fusobacterium nucleatum* [274]. Such observations pose exciting questions as to how tsRNAs produced in microbes could affect health and disease states of the mammalian host. Since one molecular consequence of tRNA fragmentation is interference with protein translation, microbiome-derived tsRNAs detectable in mammalian systems might not only be the result of the ongoing battle between microorganisms expressing killer ribotoxins but might also influence host cell physiology.

## 9. Open Questions

### 9.1. Which Genes Exactly Do Give Rise to tsRNAs?

Assigning tRNA and tsRNA sequence identity to the human genome and transcriptome remains challenging, since many tRNAs and tsRNAs map to multiple tRNA loci [275]. In addition, the observation of cell-type and even cell cycle-specific usage of particular tRNA isodecoders (tRNA genes with the same anticodon but difference in overall sequence) [276,277] complicates mapping of tRNA-derived sequence reads to genomes and therefore requires concomitant sequencing of transcriptomes. However, sequencing tRNAs and their small RNA derivatives by RT-based methodology introduces major biases, since particular modifications will hamper and even abort the activity of RT leading to extreme bias in resulting sequence output (see below). In addition, tRNA-like sequences that are not annotated as tRNA genes are abundant in the human genome, complicating the question whether a read represents expression of tRNAs or some other RNA fragment [278]. Hence, better mapping algorithms need to be developed allowing unequivocal read mapping to tsRNAs [279].

### 9.2. How to Correctly Quantify tsRNAs?

Any interested reader of the increasing body of publications reporting on tsRNA abundance and identity will notice the occurrence of specific tsRNAs in the obtained data sets derived from various model systems. Specifically, small RNAs derived from tRNA-Gly^GCC/CCC^ and tRNA-Glu^CUC^ feature often as the most abundant tsRNAs, independently of their belonging to tsRNA^S^ or tsRNA^NonS^ species. This seems at odds with the fact that stress-induced tRNA fragmentation by ANG targets many more tRNAs than just two tRNA isoacceptors. Hence, the question arises as to why some tsRNA species are better represented in high-throughput sequencing output than others?

The influence of tRNA modifications on the relative abundance of tsRNA reads in small RNA sequencing data sets has largely been ignored. Various tRNA modifications do interfere with the reverse transcription reaction. Hence, the identity of the full spectrum of tRNA-derived small RNAs remains (probably) unknown. One approach that aims at avoiding some of the biases introduced by blocked reverse transcriptase activity is called AlkB-facilitated RNA methylation sequencing (ARM-Seq) [280]. Here, enzyme-mediated demethylation removes m^1^A, m^3^C and m^1^G thereby allowing improved mapping of reads that contained such modifications and enabling the identification of tRNAs and tsRNAs that were inaccessible to reverse transcriptase-mediated cDNA synthesis. However, quantification of small RNAs by high-throughput methods, which presently requires multiple amplification steps of the original analyte sequence will never be absolute and therefore necessitates the use of orthogonal methodologies.

One such alternative might be the use of hybridization-based and customized microarrays, which does facilitate quantitative assessment of tRNAs in biological samples [243]. This approach even allows detection of single-base differences among tRNA isoacceptors [248]. While such microarrays have been already used to detect tsRNAs [281] it remains to be tested whether they allow improved quantification of tsRNAs. Importantly, since particular RNA modifications do influence Watson-Crick base pairing as well as base stacking [282,283], such effects might interfere with the hybridization of tRNAs or tsRNAs to complementary oligonucleotides on microarray platforms thereby reducing quantitative precision. However, informed design of such microarrays through, for instance, inclusion of particular probes containing compensatory nucleotide changes at positions affecting hybridization to the analyte RNA, could be exploited to address these issues systematically. Another (alternative) methodological development to quantify RNAs is represented by the sequence-specific identification of RNAs combined with their relative quantification using mass-spectrometry [284]. However, chemical modifications will affect the flight patterns of ionized RNA fragments making it necessary to include robust standards in the analysis (such as heavy isotope-labelled reference RNAs [285]) before attempting the precise quantification of different tsRNAs by mass-spectrometry. In addition, any non-sequencing-based methodology requires large amount of input material, precluding tsRNA quantification from, for instance, limited patient material. These considerations indicate that a combination of methods will likely need to be developed to quantify tsRNA abundance with sufficient sensitivity and correct stoichiometry.

### 9.3. What Is the Modification Status of Individual tsRNAs?

Since tsRNAs function in intergenerational inheritance of acquired metabolic traits was affected by RNA modifications [147,172] the question arises as to how specific tsRNAs are modified. Although one might assume that tsRNAs inherit their modification patterns from parental tRNAs, the exact modification status of individual tsRNAs, especially during the stress response remains unclear. Experimental evidence from yeast indicated that tRNA modification patterns do change during the stress response [194,286,287] suggesting that also modification patterns might differ in the resulting tsRNAs. In order to determine such pattern changes, specific tsRNAs need to be purified followed by quantitative mass spectrometry rather than ARM-sequenced. Methods to purify specific tRNAs have been established for mitochondrial tRNAs [288] and should also be applicable to tsRNA analysis.

### 9.4. Do tsRNAs Act Alone, in Pairs or Multimers?

tsRNAs appear as single molecules under almost every assay condition (i.e., Northern blotting, RNA sequencing) because initial RNA extractions are usually achieved under denaturing conditions. Are tsRNAs performing the many in vivo functions, which have been suggested as biologically significant, mechanistically as single molecules or do tsRNAs need to in interact with other molecules?

To date, there are no reports elucidating the actual stability of specific tsRNAs. Their presence in bodily fluids suggested protection against nuclease activities [78]. While tsRNAs have been localized to microvesicles and exosomes, some reports also indicated the existence of vesicle-free tsRNAs with or without associating proteins. If particular tsRNAs are indeed protein-free in vivo, the dimerization of specific tsRNAs resulting in stabilisation against degradation [55] represents an explanation for their longevity. In addition, the formation of higher-order structures such as RG4 by specific tsRNAs [152,156] points towards cellular activities that not only unwind ‘nicked tRNAs’ thereby producing individual tsRNAs in the first place but also towards activities controlling the re-association of tsRNAs into RNA-hybrids of different composition and function [55]. Methods allowing to preserve in vivo tsRNA structures are presently not available. Such technologies are certainly needed for detecting and monitoring tsRNA activity and localization in vivo.

### 9.5. Which Protein Do Associate with Individual tsRNAs?

Biologically significant tsRNA function has been reported in many diverse cellular and biological contexts. While tsRNAs might form higher-order RNA structures with each other it remains largely unclear, which proteins bind to specific tsRNAs in order to mediate their functional effectiveness.

So far, proteins associating with tsRNAs have been exclusively identified using synthetic tsRNA sequences [114,129,152,153,156,289]. However, since tsRNAs are produced from parental tRNA molecules, which carry various chemical modifications, these findings thus do not accurately reflect the in vivo situation for possible interactions between tsRNAs and their protein binders. Theoretically, chemical synthesis could also include introducing modified nucleotides at identified tsRNA positions (see above). However practically, many modified nucleotides remain commercially unavailable necessitating chemical synthesis by expert laboratories. In addition, different commercial vendors sell RNA oligonucleotides containing different amounts of modified nucleotides indicating impurities, which could change experimental outcomes, especially when aiming to define in vivo-like tsRNA-proteome interactions [290]. Hence, acquiring modified tsRNAs remains a major limitation for using defined tsRNAs for biological studies. One experimentally viable option is to biochemically purify endogenously modified tsRNAs before performing tsRNA-protein interactions studies (see above). Such experiments will not only reveal sequence- and modification dependent protein binders but will also confirm (or reject) previously published interaction studies.

## 10. Conclusions and Outlook

The growing number of reports on the astounding diversity of tsRNA-mediated functional consequences indicate that tsRNAs represent not only versatile modulators of various cellular processes (i.e., stress response, small RNA pathway function, virus infection) but also serve as conduits of information transfer across generations. Clearly, RNA modifications play a decisive role in tsRNA biogenesis and potentially also for their stability, longevity and their molecular function.

In addition, changes in tsRNA abundance and molecular manipulation of tsRNAs in various human cell culture models suggest that aberrant tsRNA production could cause systemic malfunction resulting also in human disease development. However, most of these conclusions have been made (through ‘guilt-by-association’) using classical genetic animal knockout models for tRNA processing and tRNA modification systems as well as highly selected cancer-derived cell lines. Therefore, solid functional proof using actual patient-derived material, which would allow connecting the observed changes in tsRNA abundance and identity with human syndromes is still largely lacking. Clearly, in order to convincingly connect tsRNA function to human physiology and disease development, new experimental systems need to be established. In particular, experimental systems should be avoided, which rely on selective and elevated output of protein translation machinery that often includes increased tRNA expression [250,291]. Experimental alternatives such as human organoid cultures [292] might be attractive systems combining both tissue and organ-specificity with the ease of molecular and genetic manipulation while avoiding the caveats of cancer cell lines. In addition, improved experimental designs are necessary that allow separating the loss-of-function of parental tRNAs (i.e., in genetic mutants for particular tRNA processing or modification enzymes) from the potential function of the newly produced tsRNAs. In summary, the functional versatility of the ancient and conserved tRNA molecule continues to surprise. tsRNA production as a conserved phenomenon is only one of the latest and more exciting findings. Experimental ingenuity, sufficient grant money and time will reveal the extent of what exactly tsRNAs can do, which information they transmit and if they can contribute to human disease development.

## Figures and Tables

**Figure 1 genes-09-00607-f001:**
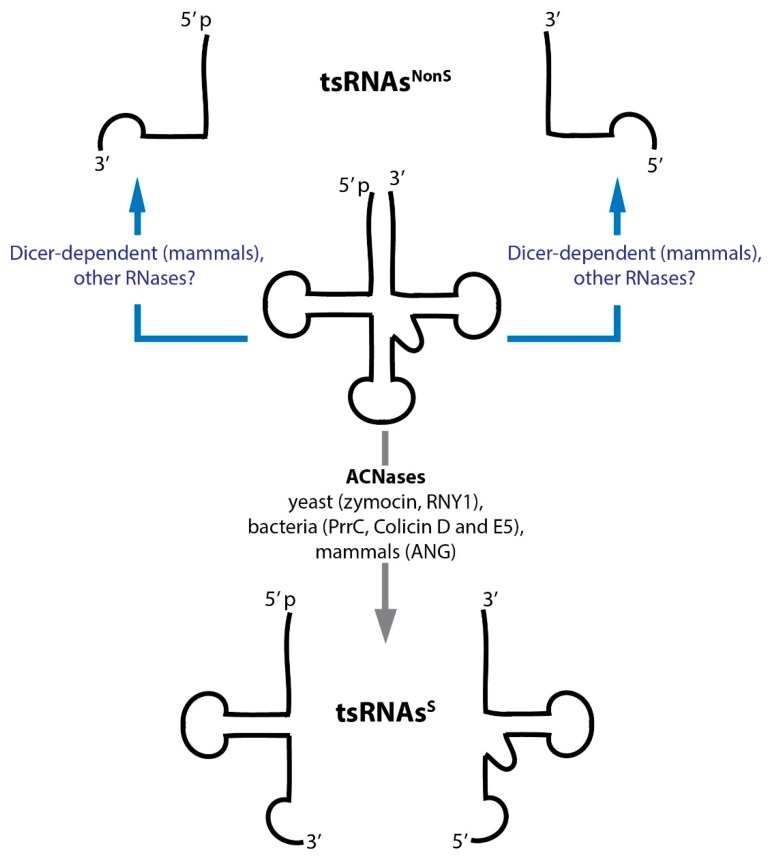
Transfer RNAs (tRNAs) give rise to various tRNA-derived small RNAs (tsRNAs). tRNAs are preferentially cleaved in open loop structures. Dicer enzymes, as well as unknown RNases cleave tRNAs in the D- or T-loops, producing short tsRNAs in a mostly stress-independent fashion (tsRNAs^NonS,^, see text). The activity of various anticodon ribonucleases (ACNases) targeting the anticodon loops produces longer tsRNAs, often during stress conditions (tsRNAs^S^, see text).

**Figure 2 genes-09-00607-f002:**
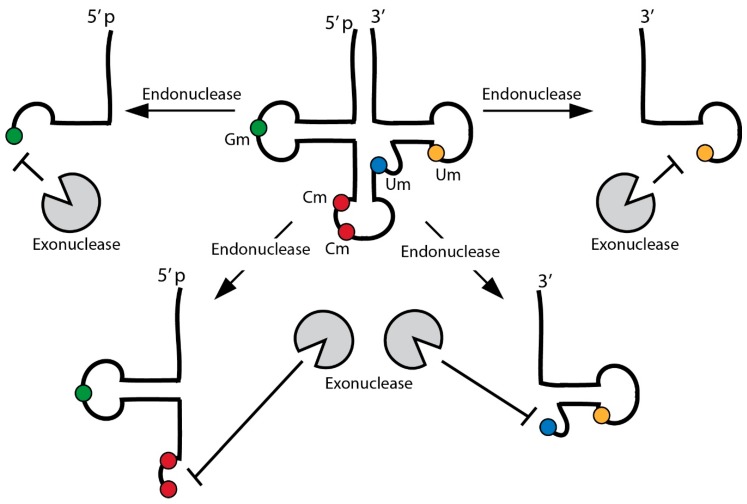
Specific tRNA modifications might influence tsRNA stability. Various tRNAs contain particular RNA modifications that inhibit the access of 3′-5′ exonucleases. In particular, the positioning of 2′-*O*-methylated nucleotides (Cm, Gm, Um) in open loop structures suggests that such modifications might stabilize produced tsRNAs against degradation.

**Figure 3 genes-09-00607-f003:**
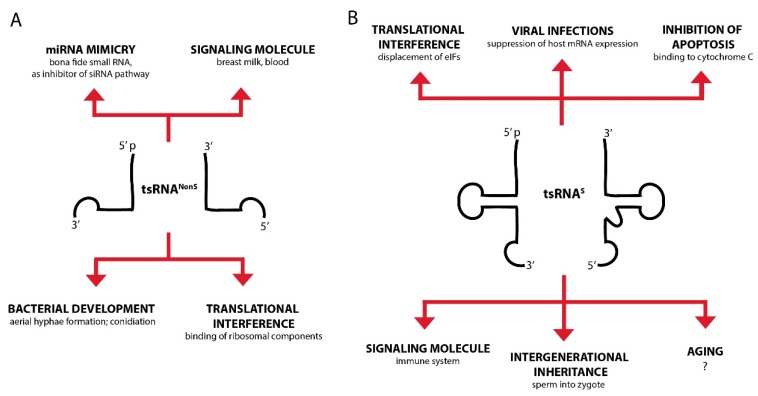
The activity of tsRNAs has been implicated in various biological processes. A selection of reported and suggested functions for tsRNAs^NonS^ (**A**) and tsRNAs^S^ (**B**) in different model organisms.

**Table 1 genes-09-00607-t001:** Experimental evidence for suggested/reported function of various tsRNAs. Table depicts experimental approaches to identify the molecular function of tsRNAs, models and model organisms, species of tsRNA and suggested/reported molecular function of individual tsRNAs.

Experimental Category ^a^	Model	Methodological Details	tsRNA	Molecular Function ^b^
4	*Mus musculus*, Mammalian cell culture	Injection and transfection of synthetic tsRNAs [134]	S and NonS	C, D
1	Mammalian cell culture	NGS [135]	NonS	B
1, 2	Mammalian cell culture (cancer)	RNA-immunoprecipitation, NGS, Northern blotting [63]	NonS	B
4	Mammalian cell culture (cancer)	tsRNA transfection [75]	S	A
1, 3	Mammalian cell culture (cancer)	Northern Blotting, NGS, qRT-PCR, RNAi knockdown of tsRNAs [76]	NonS	B
3	*Cucurbita maxima*	RT-PCR, Northern blotting [136]	S	A, B
4	Mammalian cell culture (cancer)	tsRNA transfection [137]	S	A
2, 3	Mammalian cell culture (cancer)	RNA-immunoprecipitation; Northern blotting, mRNA reporters [77]	NonS	B
2, 3	Mammalian cell culture (cancer)	mRNA reporters; RNA affinity chromatography [114]	S	A
2, 3	*Haloferax volcanii*	In vitro binding studies; mRNA reporters [117]	S	A, D
1, 2	*Tetrahymena thermophila*	RNA-immunoprecipitation, NGS [138]	NonS	D
1, 2, 3	Mammalian cell culture (cancer)	mRNA reporters; Northern blotting [139]	NonS	A
1, 2, 3	Mammalian cell culture (cancer)	Northern blotting, RNA-immunoprecipitation, NGS, mRNA reporters [140]	NonS	B
1, 4	*Drosophila melanogaster*	Northern blotting, RNA-immunoprecipitation, NGS [141]	S	D
1, 2	Mammalian cell culture	RNA-Immunoprecipitation, NGS [142]	S	C, D
4	Mammalian cell culture, zebrafish, human patient material	NGS, Northern blotting, tsRNA transfection [87]	NonS	A
1	Mammalian cell culture	NGS, RT-PCR [143]	NonS	B
1, 2	*D. melanogaster*	RNA-Immunoprecipitation, NGS [144]	NonS	B
1, 2, 3, 4	Mammalian cell culture (cancer)	Co-Immunoprecipitation, tsRNA transfection [145]	S	D
1, 2, 4	Mammalian cell culture (cancer)	NGS, RNA affinity chromatography, CLiP-Seq, transfection with mimetic or antisense oligos [129]	S	D
1, 2, 4	Mammalian cell culture (cancer), human patient material	NGS, Northern blotting, RNAi knockdown of tsRNAs [130]	S	C
1, 3	Mammalian cell culture (cancer)	RNA-Immunoprecipitation, NGS, EMSA [146]	NonS	B
1, 4	*M. musculus*	NGS, Northern blotting, small RNA microinjection, RNAi knockdown of tsRNAs [131]	S	C
1, 4	*M. musculus*	small RNA microinjection [147]	S	C
1, 4	*Arabidopsis thaliana*	synthetic tsRNA transfection [148]	NonS	B
1, 4	Mammalian cell culture (cancer), human patient material	Reporter assays [149]	NonS	C
2, 3	*H. volcanii, Saccharomyces cerevisiae*	NGS, RNAi knockdown of tsRNAs [150]	S	A, D
4	Mammalian cell culture (cancer), *M. musculus*, human patient material	in vitro binding studies; toeprinting analysis; cross-linking studies [151]	NonS	A, B
2, 3	Mammalian cell culture (cancer)	In vivo RNA cross-linking, Northern blotting, reporter assays, RNAi knockdown of tsRNAs [132]	S	A
1, 2, 3, 4	Mammalian cell culture, *M. musculus*, human patient material	tsRNA transfection, RNA affinity chromatography [152]	NonS	A, D
4	Mammalian cell culture, human patient material	iCLIP, NGS [153]	NonS	C
1, 4	*M. musculus*	Northern blotting, tsRNA transfection, RNAi knockdown of tsRNAs [154]	S	C

^a^ 1: Measuring of tsRNA abundance (Hybridisation, PCR amplification, NGS); 2: Enrichment in potential effector complexes (i.e., RNA-immunoprecipitation); 3: Indirect activity test (i.e., synthetic reporter systems); 4: Direct activity test (i.e., tsRNA by transfection or microinjection, interference with antisense oligos). ^b^ A: Interference with protein translation; B: Small RNA function; C: Signalling function; D: Protein Binder/Aptamer.

**Table 2 genes-09-00607-t002:** tRNA modifications associated with human disease. Table depicts the connection between particular tRNA modifications, their knock-out models in yeast and the resulting mutant phenotypes and published associations between human orthologous genes and human disease syndromes. The last column states when a direct link to tsRNA activity has been made in humans.

Modification	Yeast Genes *S. cerevisiae* (Sc) *S. pombe* (Sp)	Mutant Phenotype in Yeast	Human Genes	Mutant Phenotype/Disease in Humans	Direct Link to tsRNA Function?
Nuclear-encoded tRNAs					
2’-*O*-methyl	*TRM7* (Sc)	Growth defect [183]	*FTSJ1*	Intellectual disability (i.e., non-syndromic X-linked mental retardation) [183,184,185,186,187,188,189]	No
m^2^_2_G	*TRM1* (Sc)	Non-essential, temperature sensitivity [190]	*hTRM1*	Recessive cognitive disorders [191,192,193]	No
m^5^C	*TRM4* (Sc)	No effect on growth; higher sensitivity to MMS and H_2_O_2_ [194]	*NSUN2*	Autosomal-recessive intellectual disability [195,196]; Dubowitz-like syndrome [197]; Noonan-like syndrome [198]	Yes [52]
m^7^G	*TRM82* (Sc)	Growth defects [199]	*WDR4*	Indirectly linked to Down syndrome [200,201]	No
A-to-I editing	*TAD3* (Sc)	Lethal [202]	*ADAT3*	Intellectual disabilities, strabismus [203]; microcephaly and hyperactivity [204,205]	No
mcm^5^U/mcm^5^s^2^U	*ELP1* (Sc)	Delayed adaptation to changes in environment; ‘slow-start’ phenotype of spores; sensitivity to salt, temperature and 6-aza-uracil [206]	*IKBKAP*	Familial dysautonomia [207,208,209]	No
mcm^5^U/mcm^5^s^2^U	*ELP3* (Sc)	Delayed adaptation to changes in environment; ‘slow-start’ phenotype of spores; sensitivity to salt, temperature and 6-aza-uracil [210]	*ELP3*	Amyotrophic Lateral Sclerosis (ALS) [211,212]	No
mcm^5^U/mcm^5^s^2^U	*ELP4* (Sc)	Delayed adaptation to changes in environment; ‘slow-start’ phenotype of spores; sensitivity to salt, temperature and 6-aza-uracil [213]	*ELP4*	Rolandic epilepsy [214]	No
Wybuto-sine	*TRM12* (Sc)	Non-essential [215]	*TRMT12*	Breast cancer [216]	No
m^5^U	*TRM2* (Sc)	Non-essential [217]	*TRMT2A*	Breast cancer [218]	No
m^1^G	*TRM10* (Sc)	Non-essential [219]	*HRG9MTD2/TRM10A*	Colorectal cancer [220]; Diabetes type 2, intellectual disability, micro-cephaly [154,221]	Cancer: NoT2 Diabetes: Yes [154]
m^1^G/m^1^A	*TRM10* (Sc)	Non-essential [219]	*TRMT10C*	Multiple respiratory chain deficiencies, severe cardiomyopathy, mental retardation [222,223]	No
mcm^5^U/mcm^5^s^2^U	*TRM9* (Sc)	Hypersensitive to translational inhibitor at elevated temperatures [224]	*hABH8 (hALKBH8)*	Urothelial cancer [225,226]	No
mcm^5^U/mcm^5^s^2^U	*TRM9* (Sc)	Hypersensitive to the translational inhibitor paromomycin at elevated temperatures [224]	*HTRM9L*	Breast, bladder, colorectal, cervical, testicular cancer [227]; Ovarian cancer [228] Epigenetic cancer treatment [229]; Cervical cancer [230]	No
m^5^C	*PMT1* (Sp)	Non-essential [231,232]	*DNMT2*	Amyotrophic Lateral Sclerosis (ALS) [211,212]	No
Mitochondria-encoded tRNAs					
ms^2^t^6^A	n.d.	n.d.	*CDKAL1*	Diabetes type 2 [233,234,235]	No
t^5^s^2^U/nm^5^s^2^U	*MTO2/MTU1* (Sc)	Non-essential, reduced respiration [236,237]	*TRMU*	Infantile liver failure [238]; deafness [239]	No
m^5^C	n.d.	n.d.	*NSUN3*	Mitochondrial disease: developmental disability, microcephaly, failure to thrive, lactic acidosis, muscular weakness [240]	No

2′-O-methyl: 2′-O-methylribose, m^2^_2_G: N2, N2-dimethyl guanosine, m^5^C: 5-methylcytosine, m^7^G: 7-methylguanosine, A-to-I editing: adenosine-to-inosine edition, mcm^5^s^2^U: 5-methoxycarbonylmethyl-2-thiouridine, mcm^5^U: 5-methoxycarbonylmethyluridine, mcm^5^s^2^U: 5-methoxycarbonylmethyl-2-thiouridine, m^5^U: 5-methyluridine, m^1^G: 1-methylguanosine, m^1^A: 1-methyladenosine, m^2^t^6^A: N2-methyl-N6-thereonylcarbamoyladenosine, t^5^s^2^U: 5-taurinomethyl-2-thiouridine, nm^5^s^2^U: 5-aminomethyl-2-thiouridine.

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
