# Peer review of "tRNA-Derived Small RNAs: Biogenesis, Modification, Function and Potential Impact on Human Disease Development"

_genes, 2018, doi:10.3390/genes9120607_

Round 1

Reviewer 1 Report

This excellent review on tRNA fragment biogenesis and function provides a comprehensive description of this new and exciting area of the RNA field. Although this covers a broad spectrum of areas from bacteria to humans, the authors did a very good job in conceptualizing a big body of literature, including the use of nice figures and tables.

Comments on improving the manuscript:

Line 71: Cytosolic tRNAs are 73-95 nucleotides long. Mammalian mitochondrial tRNAs can be as short of 57.

Line 124: FTO has just been described also as an m1A demethylase in tRNA (PMID: 30197295). Ref. 37: Nature, not Nature publishing group.

Line 483: table 2, last two entries should indicate that these are mitochondrial tRNA modifications.

Line 642: tRNA isodecoder was first named in this 2006 paper (PMID: 17088292).

Lines 670-675: It should be mentioned that modifications also interfere with hybridization, just like single base differences, so the hybridization method would have its own issue of quantitative precision. Mass spec method has the problem of uneven flight of RNA oligos that requires a huge number of standards for precise quantitation. A very big issue for non-sequencing based method (when amplification is not used) is the requirement of very large amount of samples which should also be mentioned.

Line 10.5: In this section, it should be added that many modified nucleotides are not commercially available, and require chemical synthesis by experts. This is a major limitation for using modified tsRNAs for biological studies.

Author Response

Overall response

We thank the referees for their valuable input and detailed criticism. We have identified major points that needed addressing in the revised version:

·       We restructured sections 3 and 4. Section 4 is now included in section 3 and can be found as point 3.2.

·       We addressed the raised issues concerning microarray and mass spec approaches.

·       We included a number of additional references (suggested and non-suggested) to provide a complete summary of the current literature as well as to further clarify certain points of the manuscript.

Detailed responses

Reviewer 1:

This excellent review on tRNA fragment biogenesis and function provides a comprehensive description of this new and exciting area of the RNA field. Although this covers a broad spectrum of areas from bacteria to humans, the authors did a very good job in conceptualizing a big body of literature, including the use of nice figures and tables.

We thank the reviewer for the very positive evaluation of our manuscript.

Comments on improving the manuscript:

Line 71: Cytosolic tRNAs are 73-95 nucleotides long. Mammalian mitochondrial tRNAs can be as short of 57.

We extended our statement and included the length of mt-tRNAs in the second paragraph.

Line 124: FTO has just been described also as an m1A demethylase in tRNA (PMID: 30197295). Ref. 37: Nature, not Nature publishing group.

We thank the reviewer for drawing attention to this very recent publication by Wei et al., 2018. The reference was added to the review (reference 37). Furthermore, we fixed the error of the referencing program and changed Nature publishing group to Nature.

Line 483: table 2, last two entries should indicate that these are mitochondrial tRNA modifications.

We agree with the reviewer, that this indication will add more clarity to the table. The table is now divided into two groups: ‘nuclear-encoded tRNAs’ and ‘mitochondria-encoded tRNAs’. We also moved the human NSun3 activity on mt-tRNA-Met into this section of the table.

Line 642: tRNA isodecoder was first named in this 2006 paper (PMID: 17088292).

We thank the reviewer for drawing attention to this publication. We included the reference (Goodenbour and Pan, 2006) as reference 277.

Lines 670-675: It should be mentioned that modifications also interfere with hybridization, just like single base differences, so the hybridization method would have its own issue of quantitative precision. Mass spec method has the problem of uneven flight of RNA oligos that requires a huge number of standards for precise quantitation. A very big issue for non-sequencing based method (when amplification is not used) is the requirement of very large amount of samples which should also be mentioned.

We agree that both microarrays as well as mass-spec as alternative methods to HTP-seq-based quantification of tsRNAs pose technical problems on their own.

We included a short discussion of the raised points to this paragraph.

Regarding the microarray approaches we added references reporting that particular RNA modifications influenced Watson-Crick base pairing as well as base stacking (references 282, Roost et al., 2015 and 283, Zhou et al., 2016). However, we also suggested that custom designed probes for microarrays could be used to approach hybridization issues systematically.

Regarding the mass spectrometry approaches we mentioned that RNA modifications will affect the flight patterns of ionized RNA fragments. Therefore, we pointed out that using mass spec for quantification necessitates the use of robust standards, for instance by including heavy isotope-labelled reference RNAs. We added a representative report as reference 285 (Takoa et al., 2015).

Furthermore, we stated that non-sequencing based methods still require large amounts of sample material precluding the analyses of limited patient material.

Line 10.5: In this section, it should be added that many modified nucleotides are not commercially available, and require chemical synthesis by experts. This is a major limitation for using modified tsRNAs for biological studies.

We agree and have added this notion to the text. Furthermore, we pointed out that commercially available RNA oligonucleotides often contained different amounts of modifications, indicating impurities that could impact experimental outcomes, especially when investigating in vivo RNA-protein interactions (reference 290, Schmid et al., 2015).

We concluded that acquiring modified tsRNAs remains a major limitation for using defined tsRNAs for biological studies and suggested that biochemical purification of endogenously produced and modified tsRNAs would be a feasible alternative. (due to restructuring of the paragraphs, section 10.5 is now section 9.5 in the revised manuscript)

Reviewer 2 Report

This review by Oberbauer & Schaefer is a very thorough and timely insight in the diverse and complex world of RNAs derived from canonical tRNAs. It provides a good summary of how RNA modifications may drive differential processing of tRNAs into various species, and also refreshes and reframes the current understanding of their function and biology. This review also highlight the challenges for the field in the future.  This is particularly important given the confusion surrounding their nomenclature. I fear this issue may still persist despite the authors best efforts to unify the field. This review should be immediately published with the addition of a few minor editorial changes for clarity.

Minor editorial issues and questions

·         “4. The ‘Epitranscriptome’ & RNA Modification Systems: Reversible Or Not?” This section appears to be superfluous and is not immediately necessary to the review of tRNA modification biology. Does it deserve its own section? Could it be included in the previous section instead?

·         Line 133: should be: “raises the question”?

·         Line 395: What is meant by the phrase “test tRNA identity”

·         Line 452: This phenomenon was also observed by Cropley et al (2016; PMID: 27656407). This reference should be included.

·         Lines 550-553. Long sentence, consider revision.

·         Lines 670-676: Is it not possible that hybridization technologies for quantification purposes will also be affected by RNA modifications? Perhaps this could be mentioned.

Author Response

Overall response

We thank the referees for their valuable input and detailed criticism. We have identified major points that needed addressing in the revised version:

·       We restructured sections 3 and 4. Section 4 is now included in section 3 and can be found as point 3.2.

·       We addressed the raised issues concerning microarray and mass spec approaches.

·       We included a number of additional references (suggested and non-suggested) to provide a complete summary of the current literature as well as to further clarify certain points of the manuscript.

Detailed responses

Reviewer 2:

This review by Oberbauer & Schaefer is a very thorough and timely insight in the diverse and complex world of RNAs derived from canonical tRNAs. It provides a good summary of how RNA modifications may drive differential processing of tRNAs into various species, and also refreshes and reframes the current understanding of their function and biology. This review also highlight the challenges for the field in the future.  This is particularly important given the confusion surrounding their nomenclature. I fear this issue may still persist despite the authors best efforts to unify the field. This review should be immediately published with the addition of a few minor editorial changes for clarity.

We thank the reviewer for the positive and very supportive attitude towards our manuscript as well as for recognizing the importance of highlighting various challenges for the future of the field.

Minor editorial issues and questions

“4. The ‘Epitranscriptome’ & RNA Modification Systems: Reversible Or Not?” This section appears to be superfluous and is not immediately necessary to the review of tRNA modification biology. Does it deserve its own section? Could it be included in the previous section instead?

We restructured this section by including section 4 into section 3. Former section 4 is now included as point 3.2 in section 3. Furthermore, we renamed section 3 to: ‘tRNA Biogenesis and Function Depends on Chemical Modifications’.

Line 133: should be: “raises the question”?

Changed.

Line 395: What is meant by the phrase “test tRNA identity”

We have clarified this statement  by pointing out that tRNA loading with their corresponding amino acid is prone to error and that proofreading mechanisms exist. We included the article of Savageau and Freter (1979) as a reference 163). Furthermore, we replaced the ‘test’ with ‘proofread’.

Line 452: This phenomenon was also observed by Cropley et al (2016; PMID: 27656407). This reference should be included.

We thank the reviewer for drawing our attention to this publication of Cropley et al., 2016, which we included as reference 174.

Lines 550-553. Long sentence, consider revision.

We revised this sentence.

Lines 670-676: Is it not possible that hybridization technologies for quantification purposes will also be affected by RNA modifications? Perhaps this could be mentioned.

We agree that microarrays as alternative method to HTP-seq-based quantification of tsRNAs poses technical problems on their own.

We included a short discussion of the raised points to the respective paragraph.

Specifically, we added references reporting that particular RNA modifications influenced Watson-Crick base pairing as well as base stacking (references 282, Roost et al., 2015 and 283, Zhou et al., 2016). However, we also suggested that custom designed probes for microarrays could be used to approach hybridization issues systematically.

See also response to Reviewer 1.
